# Doping in Recreational Sport as a Risk Management Strategy

Werner Pitsch 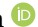

Department of Sports Science, Saarland University, 66123 Saarbruecken, Germany; we.pitsch@mx.uni-saarland.de

**Abstract:** Knowledge about the prevalence of doping in recreational sports is still limited and fragmented. The same holds true for explanations of doping prevalence rates among different groups. One of the few theoretical models that exists uses the concept of consumer capital based on Stigler and Becker's theory of rational addiction. Building on the largest study on doping in recreational sports that has ever been conducted in Europe, the FAIR+ survey, hypotheses on the differences in doping prevalence rates, by the level of participation in competitions and by the relative time spent participating in the sport are, developed. Statistical tests support the model while also drawing attention to the limitations of this theoretical explanation.

**Keywords:** performance enhancing drugs (PED); mass sport; indirect questioning techniques; Randomized Response Technique (RRT); consumer capital; rational addiction theory

## 1. Introduction

Doping in recreational sports has been studied for years, but due to the differences in methodological approaches used in the population groups being studied and in the precipitating factors of doping behavior, knowledge about the prevalence of doping as well as about its social and psychological determinants is scarce and fragmented (Frenger et al. 2016; Lentillon-Kaestner and Ohl 2011). In a recent study, researchers from different European countries addressed this divergence by conducting the first multi-national study on doping in recreational sports in Europe (Christiansen et al. 2022).

An evolving line of empirical studies since 2005 has succeeded in reliably estimating the prevalence of doping in elite sports by using indirect questioning techniques (Fincoeur et al. 2013; Fincoeur and Pitsch 2017; Pitsch et al. 2007; Pitsch and Emrich 2012; Ulrich et al. 2018; Uvacsek et al. 2011). Different variants of the Randomized Response Technique (RRT), as well as the single sample count technique were used in these studies. The results show that the prevalence of doping in elite sports ranges from between 10% and 75% and hinges primarily on the type of sport, sex and the level of athletic performance. Empirical research thus provides reliable results on the extent of doping in elite sports in different contexts, while the development of theories explaining the reasons behind the doping phenomenon is still lagging. This discrepancy was already pointed out by Pitsch and Emrich (2012), but little effort has been made since to close this gap between theoretical and empirical research.

Comparative research into recreational sports is even more scarce than it is in elite sports. To date, only two studies have examined the prevalence of doping in recreational sports using similar techniques (Frenger et al. 2016; Pitsch 2019). Interestingly, these studies found a similar pattern in the doping prevalence rates as in elite the sports, depending on the level of athletic success (Pitsch and Emrich 2012). The prevalence of doping at the most competitive levels in both elite and recreational sports was lower than at the next lower level. Moreover, the prevalence of doping in recreational sports decreased in levels below the "second tier". One explanation for this pattern is based on the concept of consumer capital within the theory of rational addiction (Stigler and Becker 1977), giving rise to a hypothesis on patterns of the prevalence of doping among different groups. This model has to date only been tested for consistency within a social scientific simulation without any further empirical testing.

This article is structured as follows: first, the prevalence of doping patterns in elite and recreational sports is briefly outlined and research hypotheses are derived. Then, indirect questioning techniques are described so the reader can independently assess the reliability of results based on the proposed method. Finally, the limitations of this research are discussed.

## 2. Theoretical Background: Doping as a Technique to Preserve Consumer Capital

The explanations for the prevalence of doping based on social science research often does not differentiate between elite and recreational sports, and empirical evidence on the differences in doping prevalence rates by discipline or by level of competition are often disregarded, which includes theories on moral (dis-) engagement, (Melzer et al. 2010), or social-cognitive theory (Barkoukis et al. 2013; Lazuras et al. 2010; Lazuras 2016; Lucidi et al. 2008; Petróczi et al. 2010). Economic theories on elite sport (Breivik 1992; Daumann 2008, 2011) usually implicitly model doping as a rational behavior, assuming that the subjectively expected utility from doping to increase competitive success ($u_{d,c}$) depends on the probability of success in the competition ($p_{s,c,d}$), the utility of competing successfully, the doping-related (monetary and moral) costs to be successful ($c_d$), the probability of detection ($p_{det}$), as well as the costs related to potential sanctioning ($c_{san}$). In short:

$$u_{d,c} = p_{s,c,d} * u_{s,c} - c_d - p_{det} * c_{san} \tag{1}$$

Yet these theories fail to explain similar prevalence patterns between different levels of competition in both recreational and elite sports despite apparent differences in both the assumed utility from being successful and the probability of detection (Berentsen 2002; Berentsen and Lengwiler 2004; Breivik 1987, 1992; Buechel et al. 2016; Tangen and Breivik 2001).

Consequently, sport science can explain the effects on either elite or recreational sports but disregards any similarities between them or provides little explanation into the effects on either category of sport that fail when empirically tested. Moreover, most theories that explain doping behavior fail to predict the prevalence of doping among different population groups in both categories of sport because psychological and economic drivers are introduced as determinants of doping behavior, while their distribution in the population group being studied remains unknown. To bridge these gaps, Pitsch (2019) developed a theory of doping based on Stigler and Becker's theory of rational addiction (Stigler and Becker 1977).

The term "consumer capital" was coined by Stigler and Becker (1977) as the central element of their theory of 'rational addiction', which could have also been referred to as "rational passion" or "rational commitment", considering that the authors described the scope of their theory using the example of "getting used to good music". In a similar vein, their publication was entitled "de gustibus non est disputandum" ('There is no accounting for taste').

The concept "consumer capital" builds on the notion that humans become used to goods which, when consumed, lead to an increased stock of human capital to continue consuming them in the future. Stigler and Becker refer to these goods as "consumer capital goods" while the proficiency to consume them is referred to as "consumer capital". The following examples clearly illustrate the utility of this concept:

- Every minute of a beginner's first skiing lesson increases his/her capacity to consume "skiing" in the future, not only in the sense of safely skiing down steep slopes but also in terms of his/her ability to small-talk about skiing at parties or to discuss skiing-related issues in a more serious setting. Moreover, the beginner's appreciation of professional skiers' performance when watching broadcasted skiing events will change.
- Elite weightlifters' potential to consume 'weightlifting' in the future increases with every training session, not only because not training would reduce their chance

of winning in competitions, but also because automatizing already acquired skills increases the likelihood of successfully competing in competitions.

The utility from consuming sport is therefore not only defined by an athlete's present consumption but is also a function of his/her past consumption, including the hours or years spent exercising and training, investments in equipment and travel, as well as the opportunity costs of training. When digging deeper into the concept of "utility from consuming sport", there is one form of utility that all athletes benefit from, namely the aesthetics of doing sport. This is an in-process and intrapsychic utility that originates in the perception of expertise when playing sport and is ranked each time an athlete succeeds in achieving an ambitious goal, be it a tricky feint in handball that results in a goal or a (double-, triple-) somersault in gymnastics. In addition, a social form of utility exists, which is the result of success in competition. Only athletes who are successful in competitions can draw on this form of positive utility from their past investments in their consumer capital. This type of utility evidently depends on the athlete's overall level of success, when considering that a world champion clearly gains more attention than a regional champion, yet recreational athletes who compete also gain positive attention in their particular social circles, such as from their family, friends and members of their sports club when they are successful at lower levels.

With regards to the above-mentioned problem of sport category-specific explanations for doping behavior, it is important for the effects to hold for both the elite and recreational sport levels. In contrast to the existing economic models on doping decisions, this notion does not limit the utility from engaging in (doped) sport of the outcome of a competition in terms of prize money and public attention but embeds the utility from sport in the individual's athletic biography. This also seems to hold for the negative utility associated with both the detection and sanctioning: penalties for doping are imposed in the form of a ban from the sport, i.e., the negative utility from such a sanction is a function of consumer capital as well.

Within this model of consumer capital, doping can be understood both in recreational as well as in elite sports as a technique to minimize the risk of losing one's utility from consuming sport.

The simulations to test this model's consistency (Pitsch 2019) is built on the notion of consumer capital assumed for differently talented individuals who played one model of sport for different lengths of time. The individual propensity for doping was derived from the probability of gaining a better position than the rankings achieved in past competitions and the probability of ranking lower. A doping decision could thus be based on increasing the utility from competing, to secure a utility that was already achieved in the past and to prevent diminished utility.

Despite being tested for consistency, this explanation of the social phenomenon "doping" has thus far only been formulated ex-post and has therefore not been explicitly tested in classical empirical social science research. Its acceptance as a "scientific explanation" is therefore questionable. Aside from deriving the theorized competition level effect from it, the simulation revealed an additional effect of the time spent participating in the sport. We therefore hypothesized:

1.  The prevalence of doping in recreational sports is highest among athletes who compete at the second highest level of performance when compared to higher and lower levels.
2.  The prevalence of doping in recreational sports increases over time in the sport the recreational athlete has been participating in.

## 3. Materials and Methods

The FAIR+ consortium (Forum for Anti-Doping in Recreational Sport) conducted a survey in eight European countries in 2021 to shed light on the prevalence, the social determinants and the psychological drivers of doping in recreational sports. Given that neither their sport nor doping is the focus of recreational athletes' everyday lives, the scope of the survey was expanded to not only cover doping but also the use of freely available

pharmaceuticals for performance enhancement, as well as sport-induced medication for purposes other than performance enhancement (e.g., mood regulation and pain reduction). Due to the sensitive nature of the studied behavior, the indirect questioning technique was used with the objective of reducing response bias.

### 3.1. Indirect Questioning Techniques

When measuring embarrassing issues in social science research, there is always a high risk of biased results owing to portrayals of social desirability. To eliminate this bias, Warner (1965, 1971) developed the RRT which has since been improved in terms of the advantages and risks of different variants of this technique (for an overview, see the meta-analysis by (Lensvelt-Mulders et al. 2005) and the recent review by Wolter (2012).

The RRT is only one among several indirect questioning techniques e.g., the unmatched count technique (Ahart and Sackett 2004), the single sample count technique (Petróczi et al. 2011), or the recently intensively studied crosswise model (Heck et al. 2018; Hoffmann and Musch 2016; Hoffmann et al. 2020; Meisters et al. 2020; Sagoe et al. 2021). The RRT was used in the FAIR+ survey because most of the evidence on doping in sport from former studies was obtained using this technique and the population group being studied (recreational athletes) made up a sufficiently large sample size, which is a precondition for using the RRT.

The FAIR+ survey used the RRT with detection for Instruction Non-Compliance (INC) (Feth et al. 2017; Clark and Desharnais 1998) in a forced response setup. The forced response method is considered one of the most efficient techniques to measure embarrassing or even threatening issues (Lensvelt-Mulders et al. 2005).

To ensure that respondents answer questions on embarrassing or even threatening issues truthfully, this technique adds random noise to the answers. This noise is produced by a randomization device, such as flipping a coin or rolling a dice during the process of answering an embarrassing question. Such randomization is directly perceived by the respondent, letting him/her experience that an embarrassing answer could result from the randomization process or from answering the embarrassing question truthfully. While respondents are perfectly safe from any inference from an answer on their characteristics or behaviors, the researcher only needs to examine the distribution of outcomes of the randomization device to arrive at conclusions based on the distribution of answers to relative rates in the population group under study. In the FAIR+ survey, we asked the respondents to select one from multiple randomly generated 5 digit figures. When asking the RRT questions, answers were randomized by referring to a certain digit of this figure, e.g., "If the second digit of your random figure is a 1, 2 or a 3, please...".

Although the RRT has proven to consistently provide more reliable answers in comparison to direct questions (Lensvelt-Mulders et al. 2005), this advantage depends on many factors which, when not properly estimated, may lead to even worse results (Krumpal and Voss 2020; Preisendörfer 2008; Wolter and Preisendörfer 2013). Most of these problems arise from the fact that respondents do not always trust the safety this technique purports or that they do not handle the randomization device properly. A technique to control this effect is the so-called cheater detection model developed by Clark and Desharnais (Clark and Desharnais 1998; Feth et al. 2017). As the RRT is often used for illegal or illegitimate issues, we will use the term "instruction non-compliance" (INC) to avoid common confusion between cheating in the sense of e.g., doping, and cheating in the sense of not answering the questions in accordance with the RRT instructions. Figure 1 presents an example of the RRT questions asked in the FAIR+ survey.

## Survey on the use of medication in recreational sport for the year 2019

|  | *If the second digit of your random number is* |  |
|---|---|---|
| * | *a 1, 2, or a 3* please answer the question on the **right side**, |  |
| * | *a 4 or a 5* please answer the question on the **left side**, |  |
| * | *otherwise* please answer the question **in the middle**. |  |
|  | Show random numbers |  |

| Does every week have 9 days? | When participating in basketball in 2019, did you **knowingly use prohibited substances or methods** to enhance your sporting performance? | Does every week have 7 days? |
|---|---|---|

Answer: ◯ Yes   ◯ No

back     Notes on answering the questions     next

**Figure 1.** Example of an RRT question on doping in recreational sport from the FAIR+ survey. For the full survey, see https://fp.socioeconomy.eu/index.php (accessed on 28 November 2022).

Ethical approval for the study, the questionnaire and methodology, including the handling and storing of data, was obtained from Saarland University (Ethikkommission der Fakultät für Empirische Humanwissenschaften und Wirtschaftswissenschaft).

*3.2. Questionnaire and Sampling*

The questionnaire was developed as an online survey. The first questions focus on socio-demographic issues and on the sport(s) the respondent plays (up to four types of sport in total), including level of competition and the time the respondent has spent participating in the sport(s). As the survey had been designed to start in 2021, questions on sport, doping and sport-induced medication use were asked retrospectively for the last pre-COVID year 2019. The ensuing questions were two RRT questions on over-the-counter medication use for performance enhancement, and the use of medication for training or competition for reasons other than performance enhancement. Additionally, we asked about doping use in up to two sports. The two sports were prioritized from up to the four sports listed by the respondent based on his/her level of competition (the highest level with the highest priority) and by the time spent participating in the sport (increasing priority with an increasing time period).

The concept of "doping" in recreational sports as addressed in the survey differs from the well-known concept of "doping" in elite sports in at least two aspects. While in elite sports, WADA defines what falls under the scope of the legal term "doping" (WADA 2020), this definition is nearly meaningless for recreational athletes because doping tests are typically not conducted in this sphere. For this reason, one cannot assume that the participants are aware of the WADA definition and the list of prohibited substances and methods (WADA n.d.). Therefore, the respondents were not asked whether they had engaged in "doping" but whether they had willingly used prohibited substances or methods to enhance their performance. This concept of "doping" differs from WADA's legal definition while it nevertheless addresses the voluntary use of substances, which the recreational athlete believes are prohibited. Additionally, the concept of "doping" in the sense of the aforementioned question differs from that of "doper" in elite sport. In elite sport, a person who violates an anti-doping rule, according to Article 2.1 to 2.11 of the World Anti-Doping Code (WADA 2020), is considered a 'doper' and could be banned from participating in organized sport, be it an athlete, a coach or a member of an elite athlete's

support staff. For recreational athletes, this understanding of being a "doper" with all of the consequences is meaningless. Therefore, the prevalence of "doping" in recreational sports reflects the relative frequency with which recreational athletes knowingly use substances to enhance their performance in a given sport, while at the same time potentially participating in a different sport without engaging in "doping".

To best approach the known concept of "doping" and "doper" as it is used in elite sport based on the FAIR+ data, only the RRT answer for doping in the sport with the highest priority was analyzed in this study, thus reducing the analysis to one sport per participant.

The FAIR+ consortium selected eight European countries for their sample (Norway, Denmark, the United Kingdom, Germany, Spain, Italy, Greece and Cyprus), covering northern, central and southern Europe. This selection was based on the home countries of the consortium members in charge of conducting the research (Denmark, Germany and Italy) and from researchers' colleagues' readiness to assist in language issues and troubleshooting. The questionnaire was initially developed in English. The questions, formulations, individual words, as well as the sequence of questions, were intensively discussed to determine the best possible phrasing. The survey was then translated by professional translators from English into six other languages (Greek for Greece and Cyprus, Danish, Norwegian, German, Italian and Spanish). For each language version, the native speaking academic partner checked the survey for congruence with the English template and for comprehensibility. Small pilots with peers and students were run for each language version,. Dissemination was primarily conducted by snowball sampling on social media platforms. This was organized by student assistants in all participating countries (except for Cyprus which was covered by the student from Greece).

The survey webpages remained active for 12 weeks from May 2021 to July 2021. After this phase, the data were checked for trustworthiness (e.g., arising from respondents who tampered with the survey by entering nonsense).

Due to the sampling procedure used, there was no opportunity to intentionally select respondents to ensure a representative sample. Consequently, the sample was heavily biased in terms of an over-representation of recreational athletes from Denmark and Spain, in particular, but also in terms of an over-representation of younger athletes. We therefore applied weighting procedures (Elliot 1991; Häder and Gabler 1997) to correct for the bias by country, sex and age based on the known distribution of the population group of recreational athletes (for details of this weighting procedure, see Christiansen et al. 2022; the population structure was derived from the European Union 2018).

### 3.3. Statistics

Responses to the RRT questions cannot be analyzed the same as responses to direct questions. RRT questions provide an estimate of the prevalence doping rate within a population group but in no way provide information about the individuals who answered the question. As prevalence rates are only meaningful concepts at the level of (sub-) populations and not at the individual level, classical statistical analyses such as t-tests and ANOVA, which build on the individual data, cannot be used. To test whether the hypotheses hold, significance tests can only be calculated to determine whether the prevalence of doping differs between groups of individuals, who, e.g., compete at different competitive levels or have spent different times participating in the sport.

In addition, using classical confidence intervals as well as statistical tests that build on the assumption of normally distributed error components in the population is strongly discouraged in RRT setups with INC detection (Frenger et al. 2016). If any of the prevalence rates that are to be estimated (honest yes, honest no or INC) is close to 0, the estimator builds on marginal solutions to account for the mathematically unnatural conditions that none of these prevalence rates may be smaller than 0 and that the sum always equals 1. In these cases, "artificially" setting one parameter to 0 leads to skewed distributions of the other estimators, thus violating the normality assumption (see Figure 2).

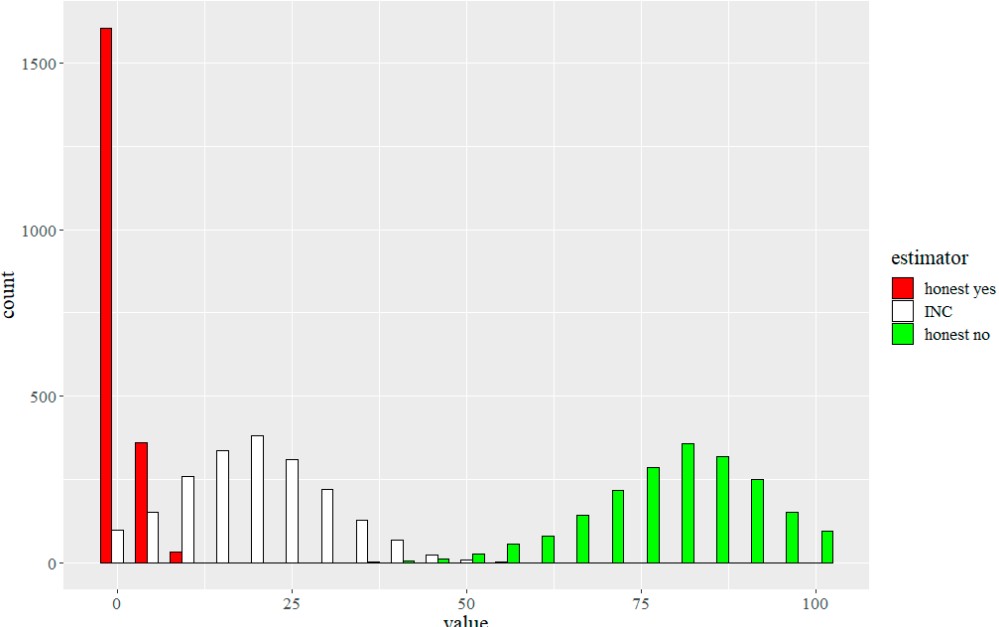

**Figure 2.** Skewed distributions of RRT estimators from 2000 bootstrap-replications for assumed true scores close to zero (honest yes) and due to the 'trade-off' of answers between INC and honest no in cases of estimators above 1 or below zero.

Confidence interval estimations, as well as hypothesis testing, were therefore conducted using non-parametric bootstrapping (Efron 1981; Efron and Tibshirani 1993). All calculations were conducted in self-developed scripts and functions in R, 4.1.3 (R Core Team 2022).

Apart from these limitations to statistical inference, RRT with INC has a unique strength. For direct questioning, the information provided by 'yes' and 'no' answers are perfectly redundant. In an RRT with INC detection, we identified three population groups, namely 'honest yes' and 'honest no' respondents as well as those who did not answer the question in accordance with the instructions. The estimate for 'honest yes' respondents is independent from that for 'honest no' respondents. Thus, not only are we able to calculate the significance of our data based on the assumption that the rate of—in this case—dopers differs between the two groups but also, and independently, whether the rate of non-dopers differs as well. Using the common significance level of 5% for both tests, this allows us to conduct a far more in-depth test of the hypotheses than the classical approach.

This double testing of the hypotheses is limited as well, however. We used the INC detection to falsely answer "no" when respondents were instructed to answer "yes" (see "NCD" in Feth et al. 2017). In this case, there is a trade-off between the INC estimator in the form of falsely answering "no" and the estimator for honest no responses. Therefore, significant differences in the honest no responses can only be interpreted as long as the INC estimator does not simultaneously differ significantly (see below).

## 4. Results

When comparing the different levels of competition participation descriptively, the results conformed to the hypothesis (Figure 3, for confidence intervals, see supplement, Table S1) both for honest yes and honest no responses. There were also considerable differences in the levels of INC. Hypothesis tests were calculated to compare the prevalence of doping at the national level (the second highest level) to the mean prevalence at the other levels of competition participation. These tests resulted in a significant difference for honest yes responders, honest no responders but also in a significant difference in the INC between the national level compared to other levels (Table 1). The difference in the estimates for honest yes responses is a clear indicator that this hypothesis holds. The

estimated prevalence of doping at the second highest level exceeds the mean prevalence at the other levels. As the trade-off between INC estimation and the estimator for honest no responses, the significant difference in honest no responses cannot unanimously be considered an indicator of the hypothesized differences.

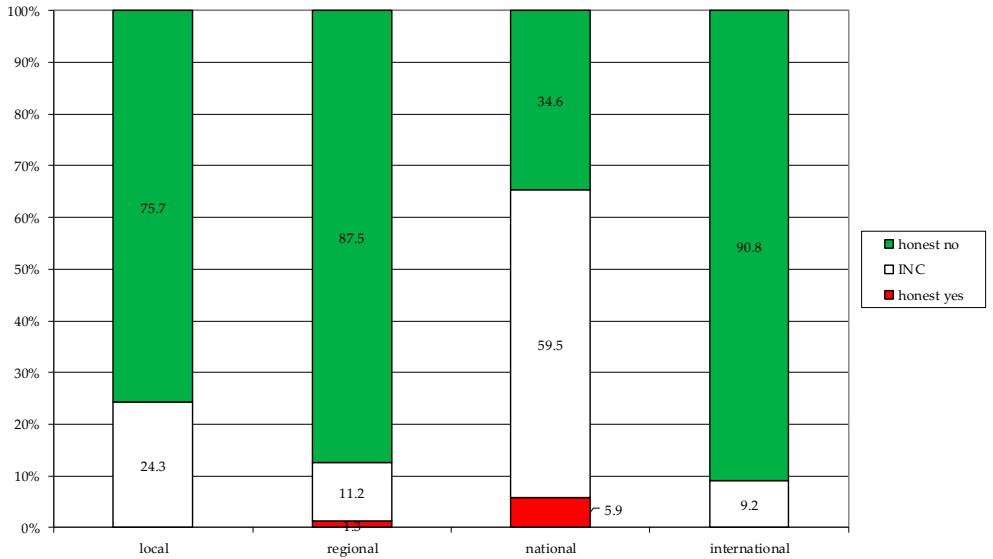

**Figure 3.** Prevalence of doping by level of competition participation.

**Table 1.** Test for significance, hypothesis 1: The prevalence of doping is at its highest at the second highest level of competition participation.

| Comparison | | Difference in Best Estimates | Confidence Interval of the Differences | |
|---|---|---|---|---|
| | | | Lower | Upper |
| National level vs. other levels n = 2862 | Honest yes | 5.9 | 0.2 | |
| | INC | 42.5 | 31.4 | |
| | Honest no | −48.4 | | −36.2 |

For hypothesis 2, the time spent participating in a sport was evaluated in terms of the relative duration in relation to the respondent's lifetime. Therefore, the time spent participating in a sport was divided by the respondent's lifetime. A descriptive analysis of the distribution of this relative duration (see Table 2) resulted in a median of 0.25 years. When calculating the median of the sample, the prevalence of the doping estimation for recreational athletes with time spent participating in the sport below the median was 0 (honest "yes"), while this estimate was 9.4 % for those with time spent participating in the sport at or above the median (Figure 4, for confidence intervals, see supplement, Table S2).

**Table 2.** Sample distribution of relative time spent participating in a sport.

| | Percentile | | | | |
|---|---|---|---|---|---|
| | 5th | 25th | 50th | 75th | 95th |
| Relative time spent participating in a sport | 0.03 | 0.10 | 0.25 | 0.55 | 0.77 |

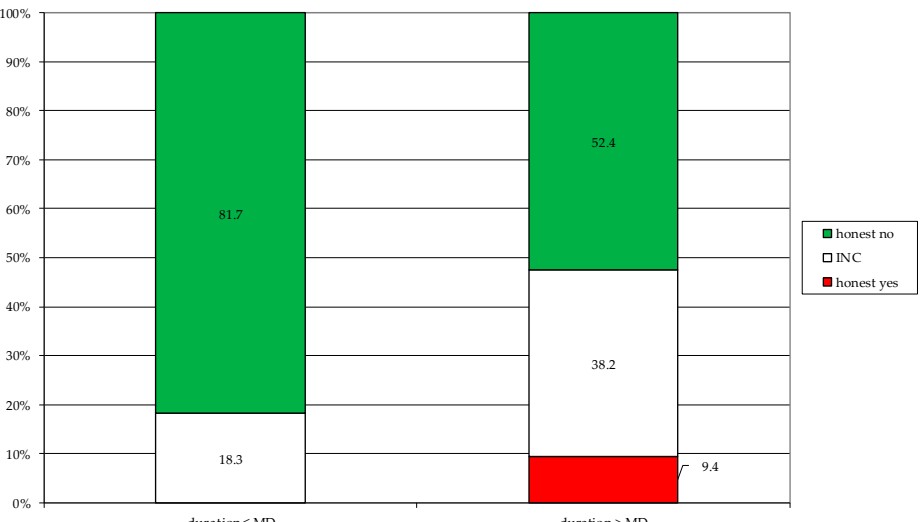

**Figure 4.** Prevalence of doping by relative time spent participating in a sport.

The significance tests (Table 3) revealed significant differences for all estimators. Likewise, hypothesis 1, which implies that the hypothesis is shown to hold for the estimation of honest yes responses but not unanimously for honest no responses due to the significant differences in INC.

**Table 3.** Significance test, hypothesis 2: prevalence of doping in recreational sports increases with time spent participating in the sport.

| Comparison | | Difference in Best Estimates | Confidence Interval of the Differences | |
|---|---|---|---|---|
| | | | Lower | Upper |
| Up to median vs. above median n = 4240 | Honest yes | 9.4 | 0.6 | |
| | INC | 19.8 | 6.8 | |
| | Honest no | −29.3 | | −12.3 |

In addition to these analyses, a descriptive analysis without an ex-ante formulated hypothesis was conducted to elaborate on the scope of the model of consumer capital to explain doping in recreational sports by comparing the prevalence of doping among competing and non-competing recreational athletes.

For non-competing athletes, the utility from playing sport ($u_{s,e}$) only consists of the in-process benefit from exercising ($u_e$), while for competing athletes, the total utility from playing sport is increased by the additional utility from competing ($u_c$):

$$\text{non-competing athletes: } u_{s,e} = u_e \qquad (2)$$
$$\text{competing athletes: } u_{s,c} = u_e + u_c$$

For $u_{s,c}$, our results reveal that it can be increased by doping, depending on the level of athletic success and on time spent participating in the sport. The effect of doping on $u_e$ can be twofold. On the one hand, doping can enhance the in-process benefit from exercising ($u_{e,d}$). The utility from performing "successfully" ($u_{s,d}$, i.d. perceiving to perform at a subjectively high level) will increase through doping while the probability to exercise "successfully" ($p_{s,e,d}$) will increase through doping as well. On the other hand, monetary but also moral costs ($c_d$) from engaging in doping will arise:

$$u_{e,d} = p_{s,e,d} * u_{s,d} - c_d \qquad (3)$$

Athletes cannot independently decide whether to engage in doping for competitive events or for exercising but decide to use (or not) doping substances or methods for both.

For competing recreational athletes, the utility of doping $u_{c,d}$ would be (by adding (1) to (3) and accounting for the fact that the costs for doping only occur once):

$$u_{c,d} = p_{s,c,d} * u_{s,c} + p_{det} * u_{san} - c_d + p_{s,e,d} * u_{s,d} \qquad (4)$$

When comparing (3) with (4), we immediately find that the costs for doping $c_d$ do not affect the different utilities from only exercising and from also competing when doped. As is the case in recreational sports, the probability of detection is practically zero because there are no doping tests; therefore, the term $p_{det} * u_{san}$ in (4) can be assumed to be equal to 0. As a result, the utility from doping for competing recreational athletes should always exceed the utility from doping of non-competing athletes and hence the prevalence of doping among competing recreational athletes is assumed to exceed the prevalence among non-competing recreational athletes. The results descriptively contradict this assumption (see Figure 5, for confidence intervals, see supplement, Table S3), but a significance test did not reveal any significant effects (Table 4).

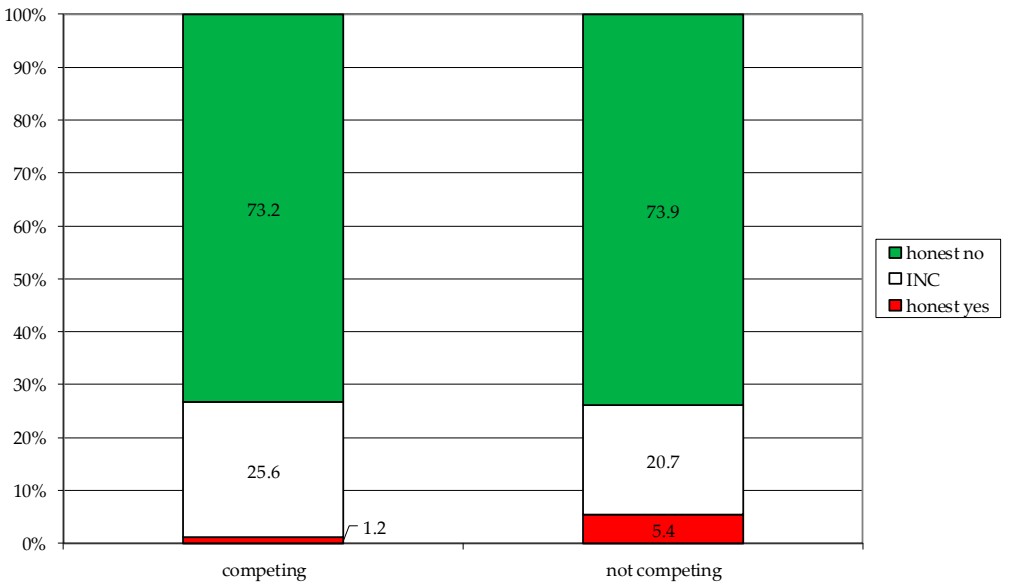

**Figure 5.** Prevalence of doping among competing and non-competing elite athletes.

**Table 4.** Ex-post test for significance for competing and non-competing athletes.

| Comparison | Difference in Best Estimates | Confidence Interval of the Differences | |
| --- | --- | --- | --- |
| | | Lower | Upper |
| Non-competing athletes vs. competing athletes n = 4193 | Honest yes   6.2 | −6.0 | |
| | INC   5.7 | −14.1 | |
| | Honest no   −11.8 | | 10.2 |

## 5. Discussion

The study was designed to test the notion that sport is a consumer capital good that increases the human capacity to play sport in the future. Based on this assumption, "doping" can be viewed as a strategy to reduce the risk of losing one's utility from playing sport. In addition to being tested for consistency with already available empirical evidence on the prevalence of doping (Pitsch 2019), this new study presents the first test of this model using newly available data.

The FAIR+ project is the largest ever empirical study on doping in recreational sports. Sampling by age, sex and country was corrected by appropriate weighting procedures. Hence, a sound database for this empirical test was available.

In hypothesis 1, the starting point for the development of the consumer capital model was addressed. In former studies on doping in elite sports (Pitsch et al. 2007; Pitsch and Emrich 2012), the prevalence of doping at the second highest level of competitive events was higher than at the highest level. In recreational sports (Frenger et al. 2016), the second highest level revealed a higher prevalence of doping when compared to the highest level but also when compared to the lower levels. The immediate in-process utility from playing sport and in training and exercising increases each time sport is consumed. This aspect does not depend on success in a competition. Other social aspects of the utility from playing sport, e.g., the gain in positive attention from relatives and friends when winning in a competition nevertheless hinges on the level of participation in competitions and on winning.

The fact that the highest prevalence of doping was not measured at the highest level of competition participation but at the second highest level was shown to depend on the relationship between time spent participating in the sport, the level of individual talent and the performance density, which differs between the different levels. The related hypothesis 1 was proven to hold for the estimate of honest yes responses, while the results for honest no responses was inconclusive. Nevertheless, support for this hypothesis is already as strong as the result of a test for significance, which could be conducted with data from a classical direct questioning survey.

Hypothesis 2 scrutinized another implication of the model that is tested here. If sport is understood as a consumer capital good, the utility from engaging in sport increases with time spent participating in the sport. The operationalization of this time as the relative duration throughout an individual's lifetime accounts for different estimations of the effort and time spent participating in the sport for differently aged recreational athletes. For the above-mentioned social aspects of the utility from engaging in sport, the risk of losing this utility can be lowered by using illegal substances or methods to increase performance. This hypothesis held for honest yes responses while honest no responses could not be interpreted conclusively.

The additionally conducted comparison between recreational athletes who compete and those who do not compete points to an important limitation. While the economic model would imply that the prevalence of doping among competing recreational athletes exceeds the prevalence among non-competing athletes, the results at least descriptively contradict this assumption. This implies that doping in recreational sports cannot simply be understood using concepts that have proven valuable in elite sports.

These results, by and large, support the model and its further development while pointing to the limited scope of its explanatory power when used in such a multifaceted domain, such as recreational sports. One promising impact of this model, beyond recreational sports, could be predicting the effects of rule amendments in terms of limitations between levels of competition, which might allow a fine-tuning of rules in a way that the probability of athletes using doping substances is reduced.

There are several limitations to this study. Most of them originate from the concept of "doping" in recreational sports, which is as multifaceted as recreational sport itself is. This renders it questionable whether the tools that have been applied in this study are appropriate for the concept under study. Nevertheless, this is a weakness of any study that attempts to open a new field of research. Another weakness originates from the survey and from the sampling techniques, which led to biased return rates and to a high level of item-non-response. The known bias was corrected by using appropriate weighting techniques. Nevertheless, the high item-non-response is unsatisfactory and leads to the problem that weightings have to be applied on a per-question basis, rendering the comparison between different questions and even between different analyses of the same question problematic. The extent to which the RRT has increased these effects can be addressed in a study that could use the newer and to date still promising Crosswise model for the same population group (Sagoe et al. 2021; Yu et al. 2008; Hoffmann et al. 2015; for a critical discussion, see also Walzenbach and Hinz 2019).

The results presented in this article should generally be understood as a first step toward scientifically exploring the concept of "doping" in recreational sports. Regarding the social impact of science, this step is overdue because anti-doping organizations already conduct anti-doping tests in this field of sport which, according to our results, science has thus far not yet really understood (Henning and Dimeo 2015).

**Funding:** This project was co-funded by the Erasmus+ Program of the European Union. Disclaimer: European Commission support for the production of this publication does not constitute an endorsement of the contents which reflects the views of the authors only, and the Commission cannot be held responsible for any use which may be made of the information contained therein.

**Supplementary Materials:** The following supporting information can be downloaded at: https://www.mdpi.com/article/10.3390/jrfm15120574/s1, Table S1: Bootstrapped CI's for hypothesis 1; level of competition; Table S2: Bootstrapped CI's for hypothesis 2, relative time spent participating in a sport. Table S3: Bootstrapped CI's for competing and non-competing athletes.

**Data Availability Statement:** Data as well as further information on data quality management and weighting procedures can be downloaded from: Pitsch, Werner, Ask V Christiansen, Monika Frenger, and Andrea Chirico. 2022. "Data and Data Management Documentation for FAIR+ Survey Data." OSF. 28 November. doi:10.17605/OSF.IO/JXZA5. R scripts and functions are available on request.

**Acknowledgments:** This article would not have been possible without the support from the other members of the technical expert group 1 in the FAIR+ project, namely Andrea Chirico, University La Sapienza, Rome, IT; Ask Vest Christansen, Aarhus University, DK and Monika Franger, Saarland University, GE. The author is also grateful to the academic partners of the survey, Vassilis Barkoukis, Aristotle University of Thessaloniki, GR; Paul Dimeo, Stirling University, UK; Jan Ove Tangen, University of South-Eastern Norway; Thomas Zandonai, the Miguel Hernández University of Elche, ES. Additionally, I want to acknowledge all members of the FAIR+ project for their contribution. For more information on FAIR+, see: https://www.europeactive.eu/fair-project (accessed on 28 November 2022).

**Conflicts of Interest:** The author declares no conflict of interest.

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
