# Peer review of "Doping in Recreational Sport as a Risk Management Strategy"

_jrfm, doi:10.3390/jrfm15120574_

Round 1

Reviewer 1 Report

The article titled “Doing in recreational sport as a risk management strategy” presents an interesting subject for further study. However, there are some points that need to be addressed as follows.

1-     A large difference is observed in INC, segregation of INC from honest-no responders in the questionnaire would probably provide more reliable results of the proposed statistical analysis (if possible).

2-     The marginal solution that is built by the estimator in the case that any prevalence rate is close to zero to avoid below zero estimation is described. This interpretation is better to be graphically illustrated with disruption in normality assumption. This means skewness in the distribution is better to be illustrated with your studied data set.

3-     The obtained results that contradicts the assumption in the this study proposed as “If winning in competition is a utility from doing sport which comes additionally to in-process-benefits from playing a sport only to those who participate in competitions, the overall prevalence of doping for those who do not compete should be lower than for those who compete because the risk in terms of the utility times the risk of detection for competing athletes is higher than for non-competing” may be further illustrated with mathematical formulation and figures to shed more light onto the obtained conclusion (if possible).

Other than the abovementioned point this research appears very interesting and answering to the abovementioned points makes it even more valuable and interesting. Therefore, a major revision is given to this research.       

Reviewer 2 Report

See attached file.

Round 2

Reviewer 1 Report

The author has provided sufficient evidence, support and clarification through explanations, mathematical models, and figures about the studied subject. I think this manuscript is ready for publication.